# Comparison of High Hydrostatic Pressure, Ultrasound, and Heat Treatments on the Quality of Strawberry–Apple–Lemon Juice Blend

**DOI:** 10.3390/foods9020218

**Published:** 2020-02-19

**Authors:** Xiaoping Feng, Zhongyu Zhou, Xiaoqiong Wang, Xiufang Bi, Yuan Ma, Yage Xing

**Affiliations:** 1Sichuan Key Laboratory of Food Bio-technology, School of Food and Bioengineering, Xihua University, Chengdu 610039, China; xiaoping-feng@nwafu.edu.cn (X.F.); zzyailijiajun@163.com (Z.Z.); wxq9695@163.com (X.W.); ymxhu@mail.xhu.edu.cn (Y.M.); xingyage1@163.com (Y.X.); 2Laboratory of Quality & Safety Risk Assessment, College of Food Science and Engineering, Northwest A & F University, Yangling 712100, China; 3Key Laboratory of Food Non-Thermal Processing, Engineering Technology Research Center of Food Non-Thermal Processing, Yibin Xihua University Research Institute, Yibin 644004, China

**Keywords:** high hydrostatic pressure, ultrasound, heat treatment, strawberry–apple–lemon juice blend

## Abstract

Changes in the microbial, physicochemical, and sensory properties of blended strawberry–apple–lemon juice were investigated to comparatively assess the influence of three processing treatments, namely high hydrostatic pressure (HHP) (500 MPa/15 min/20 °C), ultrasound (US) (376 W/10 min/35 °C), and heat treatment (HT) (86 °C/1 min) over 12 days of storage at 4 °C. The results showed that the total aerobic bacteria (TAB) counts in the HHP-, US-, and HT-treated juice blends were less than 2 log_10_ CFU/mL, the yeast and mold (Y & M) counts were less than 1.3 log_10_ CFU/mL, and the coliforms most probable number (MPN/100 mL) was less than 3 after 10 days at 4 °C. Anthocyanins were maintained by HHP, but decreased by 16% and 12% after US and HT, respectively. Total phenols increased by 18% and 7% after HHP and US, respectively, while they were maintained by the HT. Furthermore, better maintenance of total phenols, total anthocyanins, ascorbic acid, antioxidant capacity, color, and sensory values were observed in the HHP-treated juice blend stored for 10 days at 4 °C, compared to both the US- and HT-treated samples. Therefore, HHP was proposed to be a better processing technology for juice blend.

## 1. Introduction

Juice blending is one of the best methods to improve the flavor, taste, and the nutritional qualities of juice [1]. A strawberry–apple–lemon juice blend, for example, has an attractive red appearance and is rich in polyphenols, anthocyanins, ascorbic acid, and antioxidant compounds. Heat treatment (HT) is the most commonly used preservation technique to extend the shelf life of juices [2]. However, this process often induces undesirable changes that include nutrient loss, color alteration, and sensory property changes [3,4]. Non-thermal technologies have the potential to ensure health safety while maintaining the fresh-like characteristics of food products [5]. Such non-thermal processing methods, including high hydrostatic pressure (HHP) and ultrasound (US), can not only achieve the purpose of pasteurization but also maximally reserve the natural ingredients and bioactive contents of fruit juices because of their lower processing temperatures [6]. HHP is effective in inactivating microbes and enzymes while minimizing chemical reactions in food [7]. It reportedly decreases the loss of bioactive compounds (such as vitamin C, anthocyanins, and polyphenols) and maintains the physicochemical properties, including color, viscosity, and flavor, of products such as strawberry juice [8], blueberry juice [9], apple juice [10], and pomegranate juice [11]. Most studies have, however, investigated the effect on HHP on single fruit juices, and there are limited data on the use of HHP in juice blends [12]. Fernandez et al. [13] revealed that no differences were found (*p* > 0.05) in the antioxidant capacity, vitamin C, sugar, or carotene content of an orange–lemon–carrot juice blend after HHP at 500 MPa/5 min over 21 days’ storage at 4 °C. These results suggest that HHP might be a promising technology for the processing of juice blends.

Ultrasound (US), which produces sound waves of 20 kHz or more [14], can inhibit and destroy microorganisms due to the phenomenon of cavitation, in which the generation and collapse of micro bubbles result in high localized temperatures and pressure, causing disruption to the cell walls, membranes, and DNA of microorganisms [15]. Nadeem et al. [16] reported that US (525 W/5 min/15 °C) could be a suitable technology for the retention or enhancement of bioactive compounds in a carrot–grape juice blend. Gao et al. [17] demonstrated that a US-treated (20 kHz/10 min/25 °C) apple–carrot juice blend maintained (*p* > 0.05) its color, pH, titratable acid, total soluble solids, and antioxidant capacity over storage at 4 °C for 1 month, compared to the HT-treated (98 °C/3 min) samples.

Most studies have focused on the comparison of HHP or US treatments to HT in juice or juice blend processing; however, limited comparative studies have been carried out on the comparison of different non-thermal processing treatments. Therefore, in this paper, HHP, US, and HT with equivalent microbial inactivation effects on the quality of a strawberry–apple–lemon juice blend was compared and the changes in the microbial levels, pH, TSS, turbidity, instrumental color parameters, total phenols and anthocyanins, ascorbic acids, and antioxidant capacities over 10 days of storage at 4 °C were investigated. The objective of this study was to compare these treatments and ascertain the most suitable technology for strawberry–apple–lemon juice blend processing.

## 2. Materials and Methods 

### 2.1. Samples

Strawberries (chocolate strawberry, cultivar Red), apples (candy apple, cultivar Fuji), and lemons (Ulrich) were purchased from a local supermarket (Chengdu, China). 

### 2.2. Chemicals

Folin–Ciocalteu reagent (FCR) was purchased from Feijing Biotechnology Co., Ltd. (Fuzhou, China). 2,2-Diphenyl-1-picrylhydrazyl (DPPH) and 2,6-dichloro-phenol-indophenol were purchased from Yuanye Biotechnology Co., Ltd. (Shanghai, China). Plate count agar, potato dextrose agar, lauryl sulfate (lauryl tryptose) broth, and brilliant green bile lactose broth were purchased from Aoboxing Biological Technology Co., Ltd. (Beijing, China). Other chemicals were obtained from Kelong Chemical Reagent Factory (Chengdu, China).

### 2.3. Preparation of Juice Blend

Apples, strawberries, and lemons were washed thoroughly, peeled with a steel knife, and cut into slices (0.2 mm). Ascorbic acid and salt (15 g/L and 0.5 g/L, respectively) were added to the apple slices in the water (25 °C) to prevent oxidation and color change. The strawberry, apple, and lemon juices were extracted separately using a domestic fruit processor (MJ-BL25C4, Guangdong Midea Electric Appliance Manufacturing Co., Ltd., Guangdong, China). Pectinase (0.02%) was added to the strawberry juice at 45 °C for 25 min to increase juice yield [18]. The three fruit juices were then filtered separately through four layers of gauze. The fruit juice blend was prepared according to the sensory evaluations of taste, aroma, color, and consistency of 25 assessors from Xihua University, after mixing strawberry juice, apple juice, and lemon juice at ratios of 43.8%, 54.8%, and 1.4% (*v/v*), respectively. The TSS of the resulting juice blend was 7.80 ± 0.01 °Brix and the pH was 3.38 ± 0.01. The juice blend was kept at 4 °C and used within 1 h.

### 2.4. HHP, US, and HT Treatments of the Juice Blend 

Preliminary studies found that HHP at 500 MPa/15 min/20 °C, US at 376 W/10 min/35 °C, and HT at 86 °C/1 min could achieve equivalent microbial inactivation effects and meet the total aerobic bacteria (TAB) and yeast and mold (Y & M) requirements of Chinese national food safety standards for fruit and vegetable juice (GB 7101-2015) in this study’s juice blend. Therefore, the juice blends were processed under these treatment conditions and their quality attributes were then analyzed. 

For HHP treatments, according to the methods of Wang et al. [19], 50 mL screw-cap polyethylene terephthalate (PET) bottles were filled with the juice blend and placed into the vessel for processing. HHP treatments were carried out using a hydrostatic pressurization unit (HPP 600 MPa/3–5 L, Shanghai Wodi Intelligent Equipment Co., Ltd., Shanghai, China) with a capacity of 5.0 L at an ambient temperature (≈20 °C). The samples were treated at 500 MPa for 15 min at 20 °C, based on our preliminary studies. The pressurization rate was about 120 MPa/min and the depressurization was immediate (<3 s). Distilled water was used as the pressure-transmitting fluid. The treatment times reported in this study do not include the pressure-increase time and pressure-release time. 

For US treatments, the 50 mL juice blends in the PET bottles were treated by an ultrasound machine (SCIENTZ-IID, Ningbo Xinzhi Biological Polytron Technologies Inc., Ningbo, China) in a water bath (35 °C) at 376 W for 10 min. The probe had a diameter of 6 mm and the operating immersion depth was 2 cm. Pulse intervals of 2 s on and 2 s off were applied until completion. Before and after each experiment, the ultrasound probe was sterilized with 75% ethanol prior to sonication of the sample in order to avoid any contamination. After US treatment, the sample bottle was removed from the chamber and covered with a PET plastic cap to avoid microbial pollution in the air, after which the juice was diluted for further analysis.

For HT treatment, 50 mL samples were poured into beakers and placed in a water bath (DK-98-II, Tianjin Taist Instrument Co. Ltd., Tianjin, China). The samples were treated at the desired center temperature (86 °C) for 1 min, and the beakers were then removed from the hot water bath and immediately cooled in an ice bath. The heating time was 15 min and the cooling time 10 min. The HT-treated juice was aseptically transferred into 50 mL PET bottles identical to those used for the HHP and US. For all treatments, samples stored in PET bottles without any treatment were used as the control.

### 2.5. Storage Study

The treated samples were stored at 4 ± 2 °C in the dark [20]. Sample analyses were carried out after 0, 2, 4, 6, 8, 10, and 12 days’ storage.

### 2.6. Microbial Analysis 

The total plate count method [21] was used to count the viable microorganisms in the treated juice blends. Each treated and control sample was serially diluted with sterile 0.85% NaCl solution, and 1.0 mL of each dilution was plated onto duplicate plates of appropriate agar. Plate count agar was used to count the TAB after incubation at 37 °C for 48 ± 2 h, and potato dextrose agar was used to count the Y & M after incubation at 28 °C for 120 ± 2 h [21]. After incubation, the colonies were counted. Microbial data were transformed into logarithms of the number of colony-forming units (log_10_ CFU/mL).

According to Feng et al. [22], at least three consecutive dilutions are required for most probable number (MPN) analysis. Therefore, 1 mL aliquots were inoculated from each dilution into three LST tubes, which were then incubated at 36 ± 1 °C. The tube contents were examined and the reactions recorded at 24 ± 2 h for gas, then the gas-negative tubes were reincubated for an additional 24 h after being examined and the reactions recorded again at 48 ± 2 h. Confirmation tests were performed on all presumptive positive (gas) tubes. From each gassing LST or lactose broth tube, a loopful of suspension was transferred to a tube of brilliant green bile broth BGLB broth. The BGLB tubes were then incubated at 36 °C ± 1 °C and examined for gas production at 48 ± 2 h. The MPN of coliforms was calculated based on the proportion of confirmed gassing LST tubes for three consecutive dilutions.

### 2.7. pH and TSS Analysis

The pH values were measured with a digital pH meter (FZ-600T, Chengdu Century Ark Technology Co., Ltd., Chengdu, China) at 25 °C. The total soluble solids (TSS) were determined using a FAL-102 refractometer (Shenzhen Yuanhengtong Technology Co., Ltd., Shenzhen, China) at 25 °C, and the results were reported as °Brix.

### 2.8. Turbidity Analysis

For the turbidity evaluation, 30 mL of the juice blend sample was centrifuged at 3500× *g* for 15 min [23]. The absorbance of the supernatant was measured at 660 nm in a spectrophotometer (WFJ7200, Shanghai Unico Instrument Co., Ltd., Shanghai, China), using distilled water as the blank. The results for absorbance readings were correlated with the turbidity as described by Kubo et al. [24].

### 2.9. Instrumental Color Assessment 

According to the method described by Chen et al. [25], color assessment was conducted at 25 ± 2 °C using a color difference meter (WF32, Shenzhen Weifu Optoelectronics Technology Co., Ltd., Shenzhen, China). Samples of juice blend were poured into 50 mL beakers and CIE Lab values were determined in the dark. Color was expressed in *L**, *a**, and *b** values. In addition, total color differences (Δ*E**) were calculated using the following equation.
$$\Delta E^{*}=\sqrt{{\left(L^{*}-L_{0}^{*}\right)}^{2}+{\left(a^{*}-a_{0}^{*}\right)}^{2}+{\left(b^{*}-b_{0}^{*}\right)}^{2}}$$
where *L*_0_*, *a*_0_*, and *b*_0_* are the values of the control, and *L**, *a**, and *b** are the values of the treated sample.

### 2.10. Ascorbic Acid Analysis

Ascorbic acid was quantified by 2,6-dichloro-phenol-indophenol titration with minor modifications [26]. Samples of 10 mL juice blend were poured into 50 mL beakers, to which 10 mL oxalic acid solution and 4 g kaolin were added. The mixture was filtered and an aliquot of 10 mL supernatant was removed for titration with indophenol solution for 15 consecutive seconds, until a faint pink color was observed. 

Vitamin C was calculated as follows.
$$Ascorbic\;acid\left(\frac{mg}{100mL}\right)=\frac{A*B*V_{1}*100}{V_{2}*aliquot\;if\;the\;extract\;taken}$$
where *A* is volume in mL of standard dye used for titration, *B* is the weight in mg of ascorbic acid equivalent to 1 mL of indophenol solution (the dye factor), *V_1_* is the made-up volume (10 mL), and *V_2_* corresponds to the volume of sample (10 mL).

### 2.11. Total Phenol Analysis

The total phenols were determined using the Folin–Ciocalteu method as described by Cao et al. [27], with some modifications. Samples of 1 mL (previously diluted 20-fold with distilled water) were mixed with 5 mL Folin–Ciocalteu reagent (1 moL/L) and 4 mL 7.50% sodium carbonate solution, then left to sit for 1 h in the dark at room temperature. The mixture was then measured at 765 nm using a spectrophotometer (WFJ7200, Shanghai Unico Instrument Co., Ltd., China). Results were expressed as milligrams of gallic acid equivalent (GAE) per liter of juice blend (mg GAE/L), according to the calibration curve (*y* = 0.0105*x* + 0.0427), where *y* is the absorbance at 725 nm and *x* is the milligram of gallic acid equivalent (GAE) per liter.

### 2.12. Total Anthocyanin Analysis

The spectrophotometric pH differential method with minor modifications [28] was used to quantify anthocyanins. Two dilutions of the same sample were prepared using 0.025 M potassium chloride solution and 0.4 M sodium acetate solution, adjusted to pH 1.0 and 4.5 with HCl, respectively. The absorbance of each dilution was measured at 510 and 700 nm against a distilled water blank using a spectrophotometer (WFJ7200, Shanghai Unico Instrument Co., Ltd., Shanghai, China). Anthocyanin content was calculated using the following equation.
(1)$$Anthocyanin\;pigment\;(mg\;/L)\;=\frac{{A\times M}^{w}{\times DF\times 10}^{3}}{\varepsilon \times L}$$
where *A* = (A_510_ − A_700_) pH_1.0_ − (A_510_ − A_700_) pH_4.5_, MW is the molecular weight of cyanidin-3-glucoside (449.2 g/moL), *DF* is the dilution factor, *ε* is the molar extinction coefficient (26, 900 L/cm/moL), and *L* is the path length (1 cm). Total anthocyanin content was reported as milligrams of anthocyanins per liter of sample (mg/L).

### 2.13. Determination of Antioxidant Capacity

The scavenging activity of the juice blend was measured by DPPH (2,2-diphenyl-1-picrylhydrazyl) radical quenching [29]. DPPH solutions of 4 mL (0.2 mM in absolute ethanol) were added to 4 mL samples (previously diluted 20-fold with distilled water) of the juice, after which they were incubated in the dark for 30 min at room temperature (25 °C). The same procedure was conducted for the blank but absolute ethanol was used instead of the sample solution. Decreases in the absorbance were measured at 517 nm using a spectrophotometer (WFJ7200, Shanghai Unico Instrument Co., Ltd., China). The DPPH radical scavenging activity was calculated as
$${DPPH\;radical\;scavenging\;activity\;(\%)\;=\;[(A}_{0}{-A}_{1}{)/A}_{0}\times 100]$$
where *A_0_* is the absorbance of the control, and *A_1_* is the absorbance of the extracts.

### 2.14. Sensory Analyses 

The juice blends were evaluated on the Days 0 and 10 of storage [20]. The juices were subjected to sensory evaluation by 25 trained assessors from Xihua University. Samples stored in a refrigerator at 4 °C were presented to the assessors in transparent plastic cups for evaluation in the panel booths at the university sensory laboratory, which conforms to the ISO (1988) international standard. The coded samples were presented to the sensory panelists randomly to prevent any flavor carryover effects, and they were asked to evaluate the samples from four aspects (aroma, color, taste, and consistency) as listed in Appendix A according to Kaya et al. [30]. The scores were collected and the average values and total scores were calculated. 

### 2.15. Statistical Analysis 

All experiments were performed in triplicate. The data were analyzed using the Statistical Program for Social Sciences (SPSS 12.0, Chicago, IL, USA) software for analysis of variance. The Duncan test was used for multiple comparisons. The data chart was carried out using Microcal Origin 8.0 (Microcal Software, Inc., Northampton, MA, USA). The significance was established at *p* < 0.05.

## 3. Results and Discussion

### 3.1. Microbial Counts after Storage in Juice Blends Treated by HHP, US, and HT 

The results presented in Figure 1 and Table 1 show the changes in the TAB, Y & M, and coliform counts in the juice blends treated by HHP, US, and HT both immediately and after 12 days of storage at 4 °C. The initial counts of TAB, Y & M, and coliforms in the untreated juice blend were measured as 4.19, 4.21 log_10_ CFU/mL (Figure 1), and 120 MPN/100 mL (Table 1), respectively. HHP, US, and HT all caused a decrease (*p* < 0.05) in the TAB, Y & M, and coliform counts in the juice blend, and the inactivation effects of the different treatments were similar. After three treatments, the TAB and Y & M in the juice blend were below 2.0 log_10_ CFU/mL and 1.3 log_10_ CFU/mL, respectively. 

It was found that the microbial counts in the HHP-, US-, and HT-treated juice blends increased slightly over the 12 days of storage at 4 °C (Figure 1); however, the Y & M count of the HT-treated juice blend was higher than 20 CFU/mL on the 12th day, which was therefore considered to be the acceptance limit in determining the end of the juice blend’s shelf life. After 10 days of storage at 4 °C, the TAB in the juice blend was below 100 CFU/mL, the Y & M in the juice blend was below 20 CFU/mL and the levels of coliforms detected by MPN in all treated samples were below 3 MPN/100 mL. These results showed that all three of these treatments could ensure the microbial safety of this juice blend for 10 days at 4 °C. The findings concurred with those of Varela-Santos et al. [11], who found that HHP (350–550 MPa for 30, 90 and 150 s) was sufficient to keep microbial populations of pomegranate juice below the detection limit (<1.0 log_10_ CFU/mL) for 35 days at 4 °C. Kaya et al. [30] found that lemon-melon juice blends treated by HT (72 °C, 71 s) exhibited no microbial growth for 30 days at 4 °C. Lavinas et al. [31] found no microorganism growth (<1.0 log_10_ CFU/mL) in cashew apple juice treated with HHP at 350 MPa for 7 min or 400 MPa for 3 min during storage for 8 weeks at 4 °C. Martinez-Flores et al. [32] reported that the TAB count in carrot juice treated by US (24 kHz, 10 min, 58 °C) was only 2.02-log for 20 days at 4 °C. Since the inactivation effects of HHP, US, and HT on the juice blend microbes were similar, the sensory and physicochemical qualities after storage at 4 °C were compared to ascertain the most suitable processing treatment.

### 3.2. pH, Total Soluble Solids (TSS), and Turbidity after Storage in Juice Blends Treated by HHP, US, and HT

The variations in the pH, TSS, and turbidity in the juice blends treated by HHP, US, and HT after 12 days of storage at 4 °C are shown in Table 2. The initial pH value of the juice blend was 3.38. This value was not changed (*p* > 0.05) by either the HHP or the HT; however, it was increased (*p* < 0.05) to 3.43 by the US. This increase in pH value after the US treatment could be attributed to the new chemical compounds in the juice created the ultrasound action [33]. After 10 days’ storage at 4 °C, the pH values in the US- and HT-treated juice blends were decreased (*p* < 0.05), while they remained unchanged (*p* > 0.05) in the HHP-treated juice blends. After 10 days at 4 °C, pH values in the HHP-, US-, and HT-treated juice blends reached 3.37, 3.26, and 3.37, respectively. Decreases in the pH values during the cold storage period might have been due to the activity of acid-producing bacteria such as Alicyclobacillus acidoterrestris [34].

The TSS of the juice blend was not changed (*p* > 0.05) by either the HHP, US, or HT treatments, and was well maintained for 10 days at 4 °C. Previous studies have shown that HHP can maintain the TSS of apple juice [10] and of pomegranate juice [11], and also that US would not alter the TSS of orange juice [35]. 

The turbidity of juice blends was not changed (*p* > 0.05) by HHP, but was decreased (*p* < 0.05) by HT and increased (*p* < 0.05) by US. The turbidity increased by 195% after US, while it was decreased by 74% after HT in comparison to the control. The increase in the turbidity of US-treated juices could be attributed to the high pressure gradient exerted by cavitation, which might break larger molecules into smaller ones and, ultimately, thoroughly homogenizes the juice [6]. During the 10 days of storage at 4 °C, turbidity in the HHP- and US-treated juice blends decreased (*p* < 0.05), but it increased (*p* < 0.05) in the HT-treated products. Turbidity in the HHP- and US-treated juice blends was reduced by 55% and 95%, respectively, while it increased by 240% in the HT condition after 10 days at 4 °C. Cao et al. [36] found that the turbidity of cloudy strawberry juices treated by HHP (600 MPa, 4 min) during storage at 4 °C was reduced with increased storage time. Tiwari et al. [37] found that the turbidity of orange juice treated by US (40%, 70%, and 100%; 2, 6, and 10 min) during storage at 10 °C was decreased with an increase in storage time. Decreases in turbidity might be due to a loss of the viscosity, as well as to reduced stability in the juice system caused by the precipitation of bigger pulp particles and the polymerization of phenolic compounds and proteins [36].

### 3.3. Total Phenols, Total Anthocyanins, and Ascorbic Acid after Storage in Juice Blends Treated by HHP, US, and HT 

As shown in Figure 2A, the initial concentration of total phenols in the untreated juice blend was 877.69 mg GAE/L. Total phenol content in the juice blends was increased (*p* < 0.05) by 18% and 7% after HHP and US, respectively; however, it was slightly decreased (*p* > 0.05) by HT compared with the control. The increase in the phenolic compounds of US-treated juice blend could be attributed to cavitation, which caused breakage in cell walls due to the sudden change in pressure induced by bubble implosions and, thus, the release of the bound forms of these compounds [38]. Similarly, HHP might improve the extraction efficiency of these compounds by breaking the cell wall through enhanced pressure, thereby increasing their availability in the juice [39]. After the initial 2 days of storage, total phenols in the HHP- and US-treated juice blends were increased (*p* < 0.05); however, after 10 days of storage at 4 °C, the content of total phenols in all treated juice blends decreased (*p* < 0.05). Furthermore, after 10 days of storage at 4 °C, total phenols in the HHP-, US-, and HT-treated juice blends were 725, 766, and 613 mg GAE/L, respectively. Similarly, Varela-Santos et al. [11] reported that the phenolic content in pomegranate juice treated at 450–550 MPa for 30–150 s showed an increase in the first 3 days of storage, but decreased after 5 days. In this study, total phenols were possibly degraded by oxidation degradation and the polymerization of phenolic compounds with proteins in the juices [27].

As shown in Figure 2B, the anthocyanin content in the untreated juice blend was 10.95 mg/L. Compared with the control, anthocyanin content in the juice blend was not influenced (*p* > 0.05) by HHP, but was decreased (*p* < 0.05) by 12% and 16% after US and HT, respectively. This result indicated that HHP treatment had few undesired effects on the anthocyanin content of the juice blend. Cao et al. [27], similarly, reported that the total monomeric anthocyanins in strawberry juice exhibited no change (*p* > 0.05) with different treatment times (5, 10, 15, 20, and 25 min) at 400, 500, and 600 MPa. Degradation of juice blend anthocyanins by US could be attributable to the hydroxyl radicals produced by cavitation, which could open rings and thus induce the formation of chalcone [40]. Dubrovi et al. [41] found that total anthocyanin content in thermally treated strawberry juice (85 °C /2 min) was reduced by 0.7%–4.4% compared with sonicated strawberry juice (600 W/3, 6, 9 min/25 °C). Degradation of anthocyanins during heating could be related to their decomposition into a chalcone structure and further transformation into a coumarin glucoside derivative with loss of the B-ring [42]. In this study, anthocyanin content in all treated juice blends decreased as storage time increased to 10 days at 4 °C. After 10 days of storage at 4 °C, the anthocyanin contents in the HHP-, US-, and HT-treated juice blends were 8.90, 6.09, and 8.9 mg/L, respectively. These results implied that HHP had better capacity to retain anthocyanin content for 10 days of storage compared with the US and HT treatments. Anthocyanins have been previously reported to be unstable at high temperatures during processing or storage [43]. Loss of anthocyanins might also occur due to oxidation, as well as the condensation of anthocyanin pigments with phenolic compounds [44]. 

As shown in Figure 2C, the ascorbic acid (AA) content of the untreated juice blend was 2.78 mg/100 mL. Compared with the control, AA content in the juice blends were decreased (*p* < 0.05) by 9%, 11%, and 23% after HHP, US, and HT, respectively. HHP and US processing were generally regarded to be less destructive to AA than thermal processing [45,46]. This result was in agreement with previous observations [47], which found that AA content was higher (*p* < 0.05) for HHP-treated strawberry juice and blackberry juice (400, 500, 600 MPa/15 min/10–30 °C) compared to that of HT-treated samples (70 °C/2 min). Here, the AA contents in all treated juice blends decreased (*p* < 0.05) as storage time increased. After 10 days of storage at 4 °C, the content of AA in the HHP-, US-, and HT-treated juice blends was 1.44, 1.4, and 0.9 mg/100 mL, respectively. Dede et al. [48] found that HHP applications (250 MPa, 35 °C, 15 min) provided better AA values in tomato and carrot juices than HT (80 °C, 1 min) over 30 days at 4 °C and 25 °C. Zenker et al. [49] observed a higher AA retention in sonicated orange juice as compared to thermally processed samples (75 °C/1–30 s) over 49 days at 4 °C. During 10 days of storage, a strong deterioration in AA content was observed for all treatments. This deterioration could be explained first by the oxidation of AA to dehydroascorbic acid (DHAA), which was then further irreversibly converted into 2,3-diketogulonic acid [50,51].

### 3.4. Antioxidant Capacity after Storage of Juice Blends Treated by HHP, US, and HT 

Changes in the antioxidant capacity of juice blends immediately after treatment and after storage are shown in Figure 2D. Compared with that of the untreated juice blends, antioxidant activity in the HHP- and US-treated samples increased (*p* < 0.05) by 2.7% and 2.4%, respectively, but was maintained in the HT-treated ones. Similarly, Varela-Santos et al. [11] found that HHP-treated pomegranate juices (550 MPa for 30–150 s) exhibited higher (*p* < 0.05) antioxidant capacity than untreated samples. This increase in antioxidant capacity could be due to better extractability of antioxidant components [42]. Martinez-Flores et al. [32] found that the antioxidant activity of fresh carrot juice treated by US (24 kHz/10 min/58 °C) was increased (*p* < 0.05) compared with that of untreated juice. The increase of antioxidant capacity in the US-treated juice blend could be directly attributable to ultrasonically induced cavitation [52].

Over the 10 days of storage at 4 °C, the fluctuations in the antioxidant capacity of HHP- and US-treated juice blends were similar to that of the total phenols, in that they increased after the first 2 days of storage and then decreased in the following days leading up to Day 10. The antioxidant capacity in the HT-treated juice blend decreased (*p* < 0.05) over the 10 days at 4 °C. The antioxidant capacities in the HHP-, US-, and HT-treated juice blends were 91.3%, 92.9%, and 89.2%, respectively, after 10 days of storage at 4 °C. Generally, antioxidant activity was related to the total phenol, anthocyanin, and AA contents of fruits and vegetables. This study found a positive (*p* < 0.01) correlation with high coefficients between the total phenols and antioxidant capacities of the HHP- (*r =* 0.971) and HT-treated (*r =* 0.961) juice blends after 10 days at 4 °C. Furthermore, the antioxidant capacity of the US-treated juice blend (*p* < 0.05) after 10 days of storage at 4 °C was also correlated to its total phenols (*r =* 0.854) (Table 3). Similarly, Cao et al. [36] found high correlations between the total phenols and antioxidant activity (*r =* 0.886 or *r =* 0.919) of HHP-treated (600 MPa/4 min/25 °C) cloudy strawberry juice and clear juice. These high correlations suggested that total phenols played an important role in the scavenging activity of juice blends towards the DPPH • radical. Many previous studies have reported that phenolic compounds were responsible for the antioxidant capacities of fruits, and that fruits with higher phenolic contents generally showed stronger antioxidant capacities [43,53,54].

### 3.5. Color Changes after Storage in Juice Blends Treated by HHP, US, and HT 

Changes in the color of the HHP-, US-, and HT-treated juice blends are shown in Table 2. The *L** value of the US-treated juice blend did not change (*p* > 0.05) in comparison to the control; however, it was decreased (*p* < 0.05) in the HHP- and HT-treated samples, in which the liquid became darker. The *a** value of the juice blend was not influenced (*p* > 0.05) by HT; however, it was increased (*p* < 0.05) by HHP and decreased (*p* < 0.05) by US; the juice blend became more red after HHP but less red after US. There was no difference (*p* > 0.05) in the *b** values of the untreated, the HHP-, US-, and HT-treated juice blends immediately after the treatments. The decrease in *L** value marked browning in the HHP- and HT-treated juice blends, probably due to the release of intracellular compounds that made the juice slightly darker [55]. Patras et al. [42] found that HHP-treated (400, 500, 600 MPa/15 min/10–30 °C) strawberry and blackberry purées also had higher (*p* < 0.05) *a** values compared to those of the HT-treated (70 °C/2 min) samples, whereas the *a** values for both treated purées were lower (*p* < 0.05) than those of the untreated samples. 

The *L**, *a**, and *b** values in all treated juice blends were decreased (*p* < 0.05) with increasing storage time. After 10 days of storage at 4 °C, the US-treated juice blend presented the highest *L** value, while the HHP-treated juice blend presented the highest *a** and *b** values. The total color changes (Δ*E**) of the juice blend were 1.6, 0.4, and 2.2 by HHP, US, and HT, respectively. The Δ*E** value in the HT-treated juice blend increased (*p* < 0.05) faster than in the HHP- and US-treated ones, thus indicating the deterioration of its quality. Furthermore, all treated juice blends were observed to become darker, less red, and less yellow over the 10 days at 4 °C. Similarly, Cao et al. [36] reported that *L**, *a**, and *b** values decreased (*p* < 0.05) in cloudy and clear strawberry juices over 6 months’ storage at 4 °C, while Tomadoni et al. [56] found that thermal treatment (90 °C/60 s) reduced (*p* < 0.05) the *L** value of strawberry juice. Here, the *a** values of all treated juice blends decreased (*p* < 0.05) over 10 days at 4 °C, which was in accord with the change of anthocyanins. Moreover, the accordance between *a** and anthocyanins is evidence that the primary red pigment in the juice blend was that of anthocyanins, the destruction of which caused the decrease of the *a** values in all the treated juice blends. Typically, increased color degradation was associated with thermal processing as it enhanced the formation of the degradation products that affected color [57]. The decrease in the color parameters of the US-treated juice blend could be attributed to the lower polyphenol oxidase activity and the lower availability of oxygen in sonicated samples, as ultrasound waves also promoted liquid degasification [55]. It was for this reason that the HHP-treated juice blend presented better color stability than the US- and HT-treated samples. 

### 3.6. Sensory Evaluation

No sensory changes were noted in the taste, color, aroma, or consistency of the HHP- and US-treated juice blends compared with the control (Table 4). Barbosa-Cánovas et al. [58] reported that the pressure levels generally proposed and adopted by the food industry were not sufficient to disrupt covalent bonds, therefore maintaining the color, aroma, and flavor compounds responsible for the sensory quality of food. The positive effect of ultrasound in the preservation of similar processed products compared to raw products had been attributed to the removal of oxygen in this treatment [39,59,60]. 

Taste however, decreased (*p* < 0.05) by HT in comparison to the control. This result was similar to the findings of Sentandreu et al. [61], who reported a significant decrease in the fresh taste of citrus juices heated above 70 °C. Thermal processing had often been reported to affect the flavor, color, and taste of juice [5,62,63]. 

The HHP-treated juice blend in this study maintained stable taste, aroma, color, and consistency levels over the 10 days of storage at 4 °C. In the US-treated juice blend, the aroma, color, and consistency levels were maintained after 10 days of storage; however, a decrease (*p* < 0.05) in the taste of the US-treated samples was observed by the assessors. Color, taste, aroma, and consistency levels of the HT-treated juice blend began to decrease (*p* < 0.05) after the 10th day of storage. This was mainly due to the Maillard reaction, which caused the enhancement of browning and the development of an off flavor that affected the sensorial quality of juice samples after refrigerated storage [64]. These results, therefore, indicated that the HHP-treated juice blend had more acceptable taste, aroma, color, than those treated by US and HT.

## 4. Conclusions

HHP, US, and HT processing could all effectively reduce TAB, Y & M, and coliform counts in strawberry–apple–lemon juice blends. No significant differences (*p* > 0.05) were observed in the TAB, Y & M, and coliform counts between the HHP-, US-, and HT-treated juice blends after 10 days of storage at 4 °C. The total phenols and AA were more effectively maintained in the juice blends treated by HHP and US than those treated by HT, and they exhibited a higher antioxidant capacity than HT-treated samples. The juice blend treated by US was found to have a lower content of total anthocyanins compared with the HHP-treated samples. The sensory quality of the HHP-treated juice blend was judged by the assessors to be superior to those of the US- and HT-treated juice blends after 10 days of storage at 4 °C. Therefore, this study ascertained that HHP could be a potentially useful method through which to preserve bioactive compounds and reduce quality loss in strawberry–apple–lemon juice blends during refrigerated temperature (4 °C) storage for up to as 10 days.

## Figures and Tables

**Figure 1 foods-09-00218-f001:**
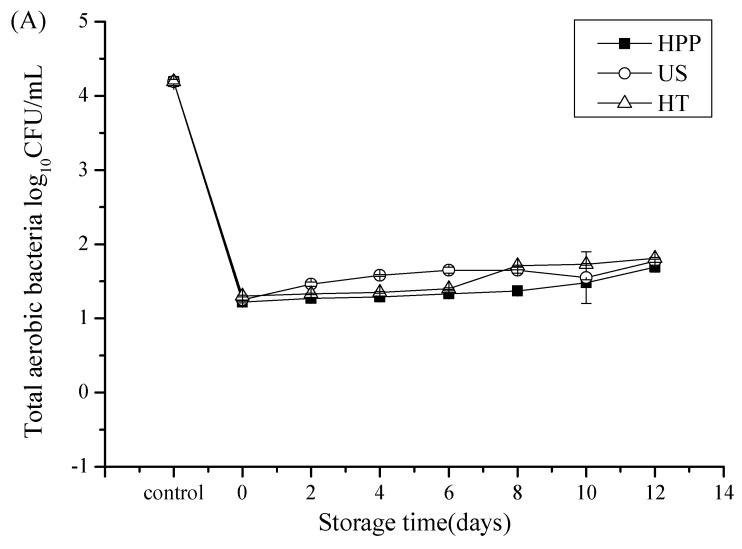
Changes in the total aerobic bacteria (**A**) and yeast and mold (**B**) counts in juice blends treated by HHP, US, and HT after 12 days of storage at 4 °C. HHP: high hydrostatic pressure (500 MPa/15 min/15 °C), US: ultrasound (376 W/10 min/35 °C), HT: heat treatment (86 °C/1 min).

**Figure 2 foods-09-00218-f002:**
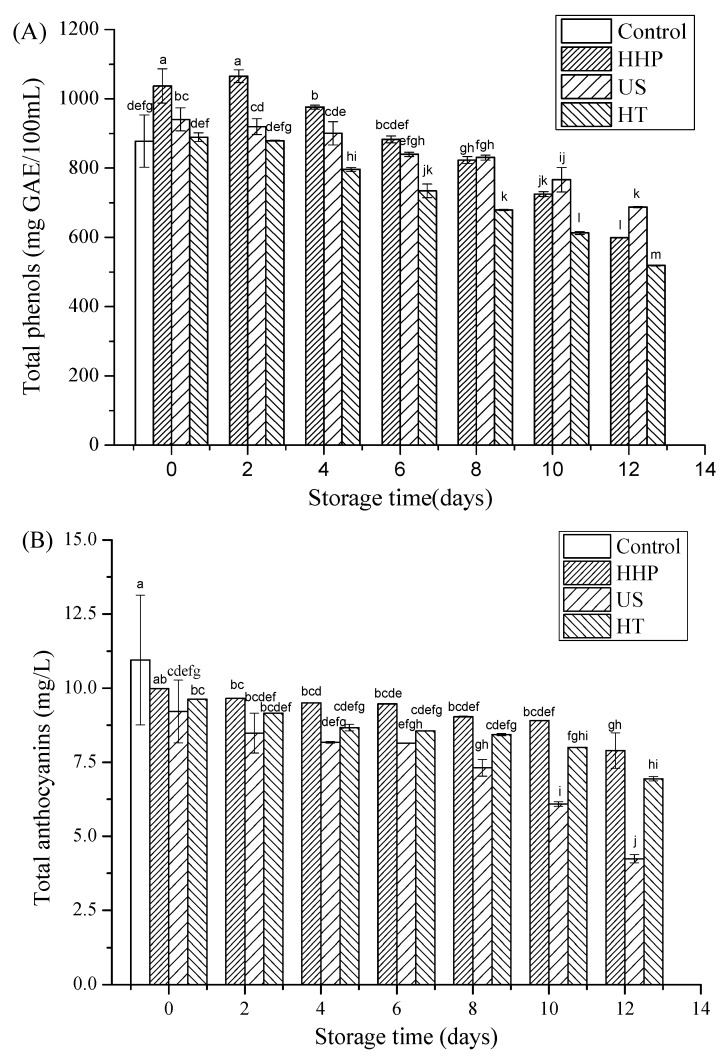
Changes in the total phenols (**A**), total anthocyanin (**B**), ascorbic acid (**C**), and antioxidant activity (**D**) in juice blends treated by HHP, US, and HT after 12 days of storage at 4 °C. HHP: high hydrostatic pressure (500 MPa/15 min/15 °C), US: ultrasound (376 W/10 min/35 °C), HT: heat treatment (86 °C/1 min). Each column represents a mean and the vertical bars indicate the standard deviation. Means with different letters within the same color columns are significantly different (Duncan test, *p* < 0.05).

**Table 1 foods-09-00218-t001:** Variations of coliforms (MPN/100 mL) in juice blends treated by HHP, US, and HT after 12 days of storage at 4 °C.

Storage Time	Treatments			
	Control	HHP	US	HT
0	120	<3	<3	<3
2	ND	<3	<3	<3
4	ND	<3	<3	<3
6	ND	<3	<3	<3
8	ND	<3	<3	<3
10	ND	<3	<3	<3
12	ND	<3	<3	<3

ND: not analyzed (the microbial count was not able to be continued due to the great growth of the microorganisms). HHP: high hydrostatic pressure (500 MPa/15 min/15 °C), US: ultrasound (376 W/10 min/35 °C), HT: heat treatment (86 °C/1 min).

**Table 2 foods-09-00218-t002:** Variations of pH, TSS, turbidity, and color parameters in juice blends treated by HHP, US, and HT after 12 days of storage at 4 °C.

Process	Storage (Days)	pH	TSS (°Brix)	Turbidity (A_660_)	Color			
					*L**	*a**	*b**	Δ*E**
Control	0	3.38 ± 0.01 ^bc^	7.80 ± 0 ^a^	0.19 ± 0.12 ^bcd^	21.59 ± 0.92 ^a^	7.29 ± 0.38 ^bc^	4.3 ± 0.20 ^abc^	0
HHP (500 MPa/15 min/15 °C)	0	3.37 ± 0.01 ^cde^	7.80 ± 0 ^a^	0.11 ± 0 ^cdefg^	20.02 ± 0 ^bc^	7.57 ± 0 ^a^	4.43 ± 0.01 ^a^	1.60 ^p^
	2	3.38 ± 0.01 ^bcd^	7.77 ± 0.06 ^a^	0.05 ± 0.06 ^efg^	19.88 ± 0.02 ^bcd^	7.50 ± 0 ^ab^	4.4 ± 0.01 ^ab^	1.73 ^o^
	4	3.38 ± 0.01 ^bcd^	7.77 ± 0.06 ^a^	0.09 ± 0 ^defg^	19.74 ± 0.18 ^cde^	7.4 ± 0.04 ^abc^	4.4 ± 0 ^ab^	1.86 ^m^
	6	3.38 ± 0.01 ^bcd^	7.80 ± 0 ^a^	0.08 ± 0 ^efg^	19.40 ± 0.07 ^cdef^	7.25 ± 0.01 ^bcd^	4.37 ± 0 ^ab^	2.02 ^l^
	8	3.37 ± 0.01 ^cde^	7.77 ± 0.06 ^a^	0.07 ± 0 ^efg^	19.13 ± 0.10 ^defg^	7.19 ± 0.01 ^bc^	4.37 ± 0.04 ^abc^	2.46 ^h^
	10	3.37 ± 0.01 ^cde^	7.80 ± 0 ^a^	0.05 ± 0 ^fg^	18.91 ± 0.06 ^fgh^	7.03 ± 0.02 ^de^	4.21 ± 0 ^abcd^	2.68 ^f^
	12	3.37 ± 0.01 ^cde^	7.80 ± 0 ^a^	0.03 ± 0 ^g^	18.45 ± 0.16 ^ghi^	6.91 ± 0.02 ^efg^	4.11 ± 0.01 ^abcd^	3.17 ^c^
US (376 W/10 min/35 °C)	0	3.43 ± 0 ^a^	7.80 ± 0 ^a^	0.56 ± 0.03 ^a^	21.59 ± 0.92 ^a^	6.91 ± 0.04 ^efg^	4.19 ± 0^a bcd^	0.40 ^u^
	2	3.35 ± 0 ^ef^	7.80 ± 0 ^a^	0.23 ± 0.09 ^b^	21.91 ± 0.71 ^a^	6.60 ± 0.06 ^hi^	4.08 ± 0.05 ^bcde^	0.79 ^t^
	4	3.34 ± 0.0 ^g^	7.80 ± 0 ^a^	0.16 ± 0.10 ^bcdef^	22.34 ± 0.10 ^a^	6.76 ± 0.03 ^fgh^	4.11 ± 0.18 ^abcd^	0.94 ^s^
	6	3.32 ± 0.02 ^e^	7.80 ± 0 ^a^	0.11 ± 0.07 ^cdefg^	22.04 ± 0.53 ^a^	6.30 ± 0.21 ^jk^	3.96 ± 0.38 ^def^	1.14 ^r^
	8	3.28 ± 0.03 ^h^	7.80 ± 0 ^a^	0.06 ± 0.05 ^efg^	20.62 ± 0.31 ^a^	6.31 ± 0.16 ^jk^	4.11 ± 0.18 ^abcd^	1.39 ^h^
	10	3.26 ± 0.01 ^hi^	7.80 ± 0 ^a^	0.03 ± 0.03 ^g^	20.16 ± 0.05 ^bc^	6.22 ± 0.05 ^k^	3.78 ± 0.31 ^ef^	2.15 ^k^
	12	3.25 ± 0 ^i^	7.80 ± 0 ^a^	0.02 ± 0 ^g^	20.01 ± 0.01 ^bc^	5.94 ± 0.14 ^l^	3.74 ± 0.01 ^f^	2.21 ^j^
HT (86 °C/1 min)	0	3.39 ± 0.01 ^b^	7.80 ± 0 ^a^	0.05 ± 0 ^fg^	19.39 ± 0.02 ^cdef^	7.42 ± 0.03 ^abc^	4.40 ± 0.1 ^ab^	2.20 ^j^
	2	3.39 ± 0.01 ^b^	7.80 ± 0 ^a^	0.07 ± 0 ^efg^	19.18 ± 0.04 ^defg^	7.21 ± 0.02 ^bc^	4.36 ± 0.01 ^ab^	2.41 ^i^
	4	3.38 ± 0.01 ^bc^	7.77 ± 0.06 ^a^	0.08 ± 0 ^efg^	19.01 ± 0.01 ^efgh^	7.01 ± 0.01 ^de^	4.31 ± 0.01 ^abc^	2.50 ^g^
	6	3.37 ± 0 ^cde^	7.80 ± 0 ^a^	0.09 ± 0 ^cdefg^	18.76 ± 0 ^fghi^	6.94 ± 0.06 ^ef^	4.29 ± 0.01 ^abcd^	2.84 ^e^
	8	3.37 ± 0.01 ^cde^	7.77 ± 0.06 ^a^	0.10 ± 0 ^efg^	18.51 ± 0.01 ^ghi^	6.87 ± 0.04 ^ef^	4.24 ± 0.01 ^abcd^	3.11 ^d^
	10	3.37 ± 0 ^cde^	7.80 ± 0 ^a^	0.17 ± 0 ^bcde^	18.01 ± 0.01 ^i^	6.67 ± 0.03 ^ghi^	4.10 ± 0.01 ^abcd^	3.64 ^a^
	12	3.36 ± 0.01 ^def^	7.80 ± 0 ^a^	0.20 ± 0 ^bc^	18.22 ± 0.02 ^hi^	6.51 ± 0.01 ^ij^	4.0 ± 0.02^c def^	3.54 ^b^

Data are presented as means ± the standard deviation. Values with different letters (a–u) in same column represent that they are significantly different from each other. HHP: high hydrostatic pressure (500 MPa/15 min/15 °C), US: ultrasound (376 W/10 min/35 °C), HT: heat treatment (86 °C/1 min).

**Table 3 foods-09-00218-t003:** Pearson correlation of total phenols, total anthocyanins, ascorbic acid, and antioxidants in juice blends treated with HHP, US, and HT.

	Antioxidant Capacity		
	HHP	US	HT
Total phenols	0.971 **	0.854 *	0.933 **
Total anthocyanins	0.943 **	0.831 *	0.917 **
Ascorbic acid	0.968 **	0.963 **	0.922 **

* Significant at 5% (*p* < 0.05). ** Significant at 1% (*p* < 0.01). HHP: high hydrostatic pressure (500 MPa/15 min/15 °C), US: ultrasound (376 W/10 min/35 °C), HT: heat treatment (86 °C/1 min).

**Table 4 foods-09-00218-t004:** Sensory scores of juice blends treated by HHP, US, and HT after 10 days of storage at 4 °C.

Storage Time	Process	Sensory Attributes				
		Taste	Aroma	Color	Consistency	Total Score
0	Untreated	33.25 ± 2.25 ^a^	24.38 ± 2.00 ^a^	11.63 ± 1.77 ^a^	12.25 ± 1.75 ^a^	81.50 ± 4.87 ^a^
0	HHP	33.25 ± 2.25 ^a^	23.50 ± 2.20 ^ab^	11.00 ± 1.6 ^a^	12.25 ± 1.75 ^a^	80.00 ± 5.37 ^a^
0	US	31.38 ± 2.82 ^ab^	22.63 ± 2.33 ^ab^	11.63 ± 1.92 ^a^	12.00 ± 1.51 ^a^	77.25 ± 3.37 ^ab^
0	HT	23.75 ± 3.78 ^c^	21.63 ± 3.16 ^ab^	10.38 ± 1.51 ^a^	11.63 ± 1.77 ^a^	67.13 ± 4.49 ^c^
10	Untreated	ND	ND	ND	ND	ND
10	HHP	31.38 ± 3.78 ^ab^	23.13 ± 2.36 ^ab^	10.38 ± 1.41 ^a^	12.00 ± 1.51 ^a^	76.88 ± 3.44 ^ab^
10	US	30.13 ± 3.76 ^b^	22.9 ± 3.00 ^ab^	11.00 ± 0.93 ^a^	11.63 ± 1.77 ^a^	73.50 ± 5.04 ^b^
10	HT	20.25 ± 0.71 ^d^	21.38 ± 2.13 ^b^	8.38 ± 0.52 ^b^	9.50 ± 0.53 ^b^	61.88 ± 3.00 ^d^

Data are presented as means ± the standard deviation. Values with different letters (a–d) in same column represent that they are significantly different from each other. ND means not analyzed. HHP: high hydrostatic pressure (500 MPa/15 min/15 °C), US: ultrasound (376 W/10 min/35 °C), HT: heat treatment (86 °C/1 min).

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
