# Peer review of "Comparison of High Hydrostatic Pressure, Ultrasound, and Heat Treatments on the Quality of Strawberry–Apple–Lemon Juice Blend"

_foods, 2020, doi:10.3390/foods9020218_

Round 1
Reviewer 1 Report
The manuscript has a large scope. I do not think that 31 pages of text are needed for original scientific research paper.
Line 94: It would be better to give a reference to a standard, recipe, or paper, why the ratio listed here was chosen.
Juice was prepared for sensory evaluation at 4 °C. I do not find this temperature suitable for the eventual detection of taste defects. I therefore recommend that consideration be given to a justification to justify this point. The temperature of the sample should be close to consumer consumption, as is usual in the home, which could be like a refrigerator temperature. But at the same time, if sensory evaluation is focused on different conservation techniques over time, the juice temperature should be such that the differences are detectable.
Line 132: Juice was stored at an interval of 2 days to 12 days. Is juice storage only considered for 12 days?
Line 168: Total color difference ΔE should be with * designation ΔE*.
The indices (a, b, c, etc.) in Table 3 do not seem right (for me) to identify statistically different groups.
Reviewer 2 Report
The manuscript in general is concise and very well written. There are but a few typing errors. The subject is modern as well as the technologies used.
The experimental design is appropriate and balanced. The resuts are presented in great detail, which sometimes is unnecessary, e.g. lines 242-251.
Why the organoleptic test was conducted in day 10 and not 12?
By comparing table, 3 and 5, in terms of consistency and turbidity, the interpretation of the data might be contradicting. Is turbidity somehow related to consistency?
Overall the manuscript can be accepted in this stage following minor revision
